# Research on Bending and Ballistic Performance of Three-Dimensional Ply-to-Ply Angle Interlock Kevlar/EP Armor Material

**DOI:** 10.3390/ma15196994

**Published:** 2022-10-09

**Authors:** Mengxiao Wang, Lin Zhong, Haijian Cao, Hongxia Chen, Xiaomei Huang

**Affiliations:** School of Textile and Clothing, Nantong University, Nantong 226007, China

**Keywords:** composite armor material, bending performance, ballistic performance, weft density, laying method

## Abstract

The three-dimensional (3D) shallow cross-bending composite material has many advantages in thickness and in-plane direction, such as high strength, high modulus, inter-layer shear strength, as well as large-area area bearing, energy absorption, etc., which has great application potential in the field of bulletproof armor. To prepare a protective material with both excellent bending performance and good ballistic performance, the effects of weft density and layering method on the bending performance and ballistic performance of three-dimensional ply-to-ply angle interlock (3DPPAI) Kevlar/EP armor materials were studied. The results showed that when the weft density of the material was 33 pieces/cm, its bending performance and ballistic resistance were the best. The 3DPPAI Kevlar/EP armor material prepared by orthogonal layup had more advantages in bending performance, and the unidirectional layup had better anti-ballistic performance. The research results will lay the foundation for structural optimization and engineering applications of such materials.

## 1. Introduction

In order to adapt to modern warfare, the requirements for protective materials in the field of ballistic protection are constantly increasing [1]. The composition of bulletproof materials has changed from simplification to diversification, such as inserts containing titanium structures [2], non-Newtonian materials [3], magnetorheological fluids [4], and the innovative development of modern new materials has brought greater improvement to bulletproof equipment space. Fiber-reinforced resin matrix composites have been widely used in the field of ballistic protection due to their lightweight, high strength, high modulus, and high specific absorption energy [5]. The research methods of ballistic response of fiber-reinforced resin matrix composites mainly include ballistic penetration tests [6,7], finite element simulations [8,9,10,11] and analytical model calculations [12,13].

The weaving structure is one of the factors affecting the bulletproof properties of fiber-reinforced resin matrix composites. At present, plain grain structural materials have been widely used in bulletproof vests [14,15,16], bulletproof helmets, and other equipment, but there are problems such as weight, easy delamination, and poor in-plane shear performance [17,18]. With the development of weaving technology, the preparation technology of 3D structural materials is gradually maturing [19,20,21]. Compared with the plain weave structure, 3D structure fundamentally overcomes the weak point of traditional laminates such as low interlayer strength and easy delamination. It also can produce high energy absorption and provide higher damage tolerance under high-speed loads [22].

Grogan et al. [23] tested vehicle armor plates with 2D and 3D woven composite backings in the URI ballistics laboratory. Analysis showed that armor panels with 3D woven backing had higher ballistic efficiency than the 2D baseline panels, with controlled delamination and fewer complete penetrations. Bandura et al. [24] studied the bulletproof performance of 2D plain woven, 3D orthogonal, and 3D angle interlock fabrics composite armor plate, and found that the increase in the ballistic limit from 2D plain woven armor to 3D orthogonal and 3D angle interlock armors was 16.44% and 20% and the degree of injury was reduced by 2.4% and 7.3%, respectively. Numerous results [25,26,27] indicate that 3D structure has excellent ballistic protection in terms of ballistic protection.

Considering the practical application of materials, the military industry has put forward dual requirements for the ballistic resistance and mechanical properties, especially the bending properties of fiber-reinforced armor materials. How to realize the light weight of armor under the premise of ensuring the anti-ballistic performance and mechanical properties of the material has become the focus and difficulty of the current research work.

The mechanical properties of high-performance fibers are the main factors affecting the ballistic penetration resistance of advanced composite materials for ballistic protection. Therefore, the key technical point to improve the ballistic resistance of advanced composites is to continuously improve the mechanical properties of high-performance fibers [28,29,30]. Although there have been breakthroughs in the mechanical properties of high-performance fibers for ballistic protection in recent years, and the ballistic resistance of high-performance fiber composites has also improved, the high production cost limits its wide application in the field of ballistic protection engineering.

Some scholars [31,32,33] improved the interfacial bonding ability between the fiber and the matrix by surface etching (Etching Modification), but there are some shortcomings such as difficult to control the surface damage of the fiber after treatment and great environmental pollution.

This paper developed a new type of 3D structure Kevlar/EP armor material. We studied the important influence of different structural parameters on the bending and ballistic properties of materials. By changing the weft density and layup method, the microstructure and overall structure of the material are changed to improve the performance of the material. The aim of this paper is to provide a new idea for the research and development of high-performance armor materials.

## 2. Experiment

### 2.1. Materials

Kevlar fiber (Yantai Taihe New Materials Co., Ltd., Yantai, China) was used to prepare three kinds of shallow cross-bending fabrics with different weft density specifications. The structure diagram is shown in Figure 1. The specific parameters of 3DPPAI Kevlar fabric are listed in Table 1.

Bisphenol A epoxy resin with a density of 1.2g/cm3 (E51, Nantong Xingchen Synthetic Materials Co., Ltd., Nantong, China); curing agent polyetheramine with a density of 0.95g/cm3 (D230, Changzhou Runxiang Chemical Co., Ltd., Changzhou, China); vacuum film, PVC hose, diversion pipe, diversion net, release cloth, sealing tape (Cixi Polymer Electronic Commerce Co., Ltd., Cixi, China).

### 2.2. Experiment Apparatus

Vacuum drying oven (DZF-6050, Changzhou China Resources Electric Co., Ltd.); Multifunctional tile cutting machine (WG-1200, Sichuan Wanguang Machinery Equipment Co., Ltd., Guanghan, China); Universal material testing machine (Instron 5969H, Instron Equipment Trading Co., Ltd. Shanghai, China); Stereo microscope (LEICASAPO, Leica Microsystems Trading Co., Ltd., Shanghai, China); Grinding and polishing machine (OMP2110, Tianjin Outlet Technology Co., Ltd. company, Tianjin, China).

### 2.3. Preparation

The epoxy resin and polyetheramine were mixed uniformly in a ratio of 4:1 and then placed in a vacuum drying oven for deaeration for 30 min to obtain a clear resin solution. The 3DPPAI fabrics laid out in a specific way were put inside a sealed bag to suction the air out of the sealed bag using a pump. Next, introduce the resin using atmospheric pressure to impregnate the fabric. Finally, the sealed bag is placed in a 75 °C oven for curing and molding, and the sealed bag is removed after two hours. Vacuum-assisted shaping is depicted in Figure 2 while fabric laying modes are presented in Figure 3.

### 2.4. Bending Performance Test

The flexural properties were obtained from three point bending tests as shown in Figure 4. The bending performance of 3DPPAI Kevlar/EP armor materials was carried out following GB/T1449-2005 standard [34]. A 2 mm/min displacement control was applied, and 5 valid data were taken for each group of samples to take the average value.

### 2.5. Ballistic Performance Test

According to the bulletproof level II standard in GA141-2010 [35], the 1954-style 7.62 mm pistol loaded with the 1951-style 7.62 mm pistol cartridge (lead core) was used to shoot the armor material 6 consecutive times at a speed of 445 ± 10 m/s. As shown in Figure 5. The shooting distance is 5 m, and six shots are fired in the order of 0°, +30°, −45°, 0°, 0°, 0°. The size of the target plate is 30 × 30 cm.

### 2.6. Destruction Morphology Observation

In order to analyze the failure mechanism, a multi-functional tile cutter was used to cut the fiber resin-based armor material along the diameter direction of the bullet hole, and each section of the target plate was polished with a polishing machine. The morphology of the bullet hole was observed by microscope.

## 3. Results and Discussion

### 3.1. Bending Performance

#### 3.1.1. Effect of Weft Density on Bending Properties

The comparative samples all have the same resin content, which had a great impact on the mechanical properties of materials [36]. The bending property of 3D armor materials with different weft densities is shown in Figure 6, and the material failure morphology is shown in Figure 7, respectively.

It can be observed that

(1) Bending performance of the 3D armor materials in weft direction is superiorly better than that in warp one [37]. When the weft density is 30, 33, 36 picks/cm, the bending strength of weft direction is 305.92, 331.09, 293.72 MPa, and the bending strength of warp direction is 172.93, 152.35, 128.27 MPa, respectively.

The warp yarn serves as the major body of the bearing when the material is bent in the warp direction. Similarly, when the material is bent in the weft direction, the weft yarn serves as the main body of the bearing. The curled warp yarns have greater axial deformability when bent in the warp direction, and the fabric exhibits excellent toughness. The warp yarns on the top layer of the material tend to wrinkle at the interlacing point following compression during the bending process, while the warp yarns on the bottom layer of the material are still undergoing tensile deformation, as shown in Figure 6. The upper and lower layers of the material cannot resist bending loads together. As a result, the flexural strength of the material is reduced.

(2) Bending strength and modulus of the 3D armor materials in the warp direction both decrease with the increase in weft density. When the weft density is 30, 33, 36 picks/cm, the bending strength of warp direction is 172.93, 152.35, 128.27MPa, and the bending modulus of warp direction is 6.41, 5.58, 4.66 GPa, respectively.

On the one hand, the arrangement of the warp and weft threads gets closer with the weft density rises, resulting in a steady reduction in the size of the pore between the yarns, finally making it harder for the epoxy resin to penetrate and creating more voids in the fabric [38,39], which affects the interface strength between the fiber and the matrix [40,41]. The voids in the material are prone to cracks and further failure of the material. The SEM morphology of cross-sectioning is depicted in Figure 8. Between the resin and the yarn, it is visible that there is an area that has not been fully infiltrated. The voids rate of 3D armor material increases from 1.46% to 1.77% to 2.21% with the increase in weft density.

On the other hand, the weft yarns are arranged in parallel without buckling, and the warp yarns are lined along the thickness direction of the structure at an angle. Warp buckling angles are different due to different degrees of warp binding in fabric systems with different weft densities [42], as shown in Figure 9. When the weft density is 30, 33, and 36 picks/cm, the warp buckling angles are 20°, 22°, and 24°, respectively. The component of the warp axis in the direction of the bending load is reduced, and damage is more likely to occur, so the bending performance is reduced. The effect of fiber orientation on bending performance has been reported by many studies [43,44,45].

(3) Bending strength of the 3D armor materials in weft direction increases at first then decreases with the increase in weft density. When the weft density is 30, 33, 36 picks/cm, the bending strength of weft direction is 305.92, 331.09, 293.72 MPa.

In comparison to the material with a fabric weft density of 30 picks/cm, the material with a fabric weft density of 33 picks/cm has a 10% increase in the volume content of weft strands. There are more fibers to share the stress and it is harder to achieve the breaking strength of the material, resulting in an 8% increase in the material’s flexural strength. However, with the increase in weft density, on the one hand, the degree of internal extrusion of the fiber intensifies, and the internal stress of the material is concentrated. On the other hand, the impregnation of resin is affected, thereby affecting the final mechanical properties [39].

#### 3.1.2. Effect of Layup Methods on Bending Properties

Different bending behavior of the 3D armor materials is indicated in Figure 10. The bending performance of the material is as follows: unidirectional ply (weft) > orthogonal ply (warp) > orthogonal ply (weft) > unidirectional ply (warp). In the unidirectional layup, bending strength and modulus along the weft direction are approximately 110% higher than those along the warp direction. In the orthogonal layup, there is little difference between the warp and weft of the material.

The reasons are analyzed as follows. 

(1) From a macroscopic perspective, the number and type of main yarns in the warp and weft directions of the material are different in the unidirectional ply, which aggravates the material difference. In the orthogonal layup, the content of warp and weft yarns in the bearing body of the material in all directions is similar, so the performance difference in the material is reduced and the material exhibits quasi-isotropy in the macroscopic sense.

(2) From the perspective of the microscopic model, according to the local failure of the material the utilization rate η represents the comprehensive bending performance of the material after changing the layering method shown in Equation (1). Due to the large difference in the warp and weft bending of materials, it is proposed that the difference rate CD expresses the difference in bending properties between the warp and weft directions of the material shown in Equation (2):η = (Q_1_ + Q_2_)/(Q_warp_*a + Q_weft_*b),(1)
CD = |Q_1_ − Q_2_|/Q_2_(2)

In the formula: η is the utilization rate; Q_1_ is the warp bending strength of the material; Q_2_ is the weft bending strength of the material; Q_warp_ is the bending strength along the meridional direction of a single fabric composite; Q_weft_ is the bending strength along the weft of a single fabric composite; a is the number of layers in the warp direction; b is the number of layers in the weft direction.

Bending coefficients of 3D armor material under different laying modes are illustrated in Figure 11. The utilization rate of the 3D armor material is like that of the unidirectional ply or the orthogonal ply, but the difference rate in the warp and weft directions of the orthogonal ply is much smaller than that of the unidirectional ply. In conclusion, the armor material prepared by orthogonal layup has more advantages in bending performance.

#### 3.1.3. Comparison of 3DPPAI and 2D Plain Weave Structure

The flexural properties of 3DPPAI armor material with a weft density of 36 picks/cm and a common armoring material with a similar surface density were compared. The specific parameters are represented in Table 2. The results are shown in Figure 12. The 3DPPAI structure has the best zonal bending performance, with a bending strength of 161.48 MPa and a bending modulus of 7.80 GPa, which are 10.45% and 4.94% greater than plain meridional and zonal bending, respectively. The warp bending performance of the 3DPPAI structure is the worst, only around 57% of that of the plain weave.

The reasons for the above phenomenon are: (1) the weft yarn is arranged in a straight line when the material of the 3DPPAI structure is bent along the weft direction, and the stress is not easy to concentrate, resulting in better structural integrity and bending performance of the material; (2) when the material of the 3DPPAI structure is bent along the weft direction, the warp yarn is lined with two layers of weft yarn along the thickness direction of the structure at a certain angle, and the buckling degree is much greater than that of the plain yarn. Therefore, the warp yarns in the 3DPPAI structure are more prone to damage, resulting in a substantial decrease in the bending properties of the material.

### 3.2. Ballistic Resistance

#### 3.2.1. Effect of Weft Density on Ballistic Resistance

According to the target shooting experiment, it is known that the fabric with the weft densities of 30, 33, and 36 picks/cm needs to be laid at least 12, 12, and 11 pieces, respectively, to prevent the impact of the second-level standard bullet in GA141-2010. The ballistic test results of 3D armor materials with different weft densities are listed in Table 3. The target plate with layers less than this standard is penetrated, and the details of the perforation are shown in Figure 13.

The back convexity and backface signature (BFS) of the target plate with three kinds of weft densities are shown in Figure 13. From the analysis of Figure 14: (1) BFS of the 1# target plate is the largest, with an average value of about 20 mm; BFS of the 2# target plate is the smallest, which is less than 10 mm, showing good trauma resistance. BFS of 3# target plate is about 12 mm; (2) the back convexity of the target plate in the three specifications is within 2–5 mm, indicating that the target plate has a large rebound after ballistic penetration, and the material has good toughness.

The back convexity and BFS of the three different weft density objectives are displayed in Figure 15. Cross-sectional views of armor materials with three weft density requirements in the 0° firing direction are presented in Figure 16.

According to Figure 14 and Figure 15: 

(1) The projectile surface of all the target plates showed penetration failure, and the fiber at the bullet hole was broken and exploded out of the plane, accompanied by fiber fibrilization;

(2) In the 1# target plate, the bullet pierced four pieces of fabric and was intercepted; in 2# and 3# target plates, the bullet pierced five pieces of fabric and was intercepted. The integrity rate of all the target plates is above 55%, and the 3DPPAI Kevlar/EP armor material exhibits high damage tolerance. This is due to the ability of interwoven yarns to significantly reduce damage by preventing shear and delamination cracks in all directions during impact;

(3) The material delamination always takes place at the reinforcement layer where the projectile stops, and the three target plates exhibit varying degrees of delamination. The reason is that in the early stage of bullet penetration, the target board consumes the kinetic energy of the bullet at the expense of the in-plane strength, and the bullet with a sharp drop in speed will continue to penetrate the target board in the incident direction or escape to the weak ring in the target board surface. When choosing between the two, the fastest way to dissipate energy is always chosen. There is no Z yarn reinforcement at the interface of the fabric layer, so it becomes the final escape channel for bullets. Due to the limitation of the weaving level of the fabric, the single fabric in the experiment has only four layers, and the final preform thickness is reached layer by layer. The delamination phenomenon will be reduced if the number of layers of a single piece of 3DPPAI fabric can meet the requirements.

(4) Among the three specifications of the target plate, the 1# target plate has the most obvious layering, 2# is the second, and the 3# only has a slight layering at the edge of the bullet. When the bullet fails to penetrate the next layer of material along the incident direction it will deflect angularly, wedging the next layer of material at a certain angle and then escaping to the junction of the fabric layup. The bullet is the easiest to wedge into the 1# target with the smallest weft density, so the 1# target is the most delaminated. As the weft density increases, the bullet is constrained by more surrounding fabric in the target plate, making it harder for the bullet to wedge around. Consequently, the bullet continues to penetrate in the direction of incidence, resulting in more layers of penetration;

(5) The reinforcing fibers at the early stage of stratification are cut neatly by shear action. The fibers deviate from the original delamination plane, forming a dorsal bulge at a later stage of delamination. Combined with Figure 14, fiber breakage occurred on the back surface of the 1# target plate. Among them, there were narrow meridional strips on the back surface of 1–4 and 6 guns, and a cross-shaped break appeared on the back surface of the 5th gun. The reason is that longitudinal waves produce axial tensile stress on layers of fibers from inside to outside, which makes the back of the target plate bulge, and the outermost fiber has the largest deformation, so the outermost fiber is more likely to be pulled off. The weft yarn is stretched out in the material and has far less strain capacity than the warp yarn, so it is the first to break. The broken weft forms a narrow strip along which the warp is drawn. The cross-shaped breach occurs because the projectile velocity of the fifth gun is the largest, the strain on the target plate is also the largest, and the warp and weft yarns at the impact point reach their breaking limits. Due to the maximum surface density of the 2# target plate, the back-projectile surface has no obvious damage except resin fragmentation. The fiber fracture and interface damage appeared on the projectile surface of the 3# target plate. Different from the 1# target plate, the 3# target plate has the largest weft density and the largest modulus of the material. Under the same impact stress, the strain of the material is the smallest. When the material is impacted by an external force, the deformation ability of the material decreases, which is not beneficial to the fiber deformation and energy absorption in the target plate, so the elastic surface of the 3# target plate has fiber damage. The interface damage is due to the maximum deformation of the outermost fiber, and the layer between the outermost fabric and the penultimate fabric is more likely to occur only by the interface adhesion.

Due to the difference in pores caused by fabric structure and the fact that the fabric content can only be controlled by adding or subtracting the number of fabric layers, the density gradient of the target plate surface is large. The above experiments can only show that the 3# target plate is better than the 1# target plate, and the 2# target plate and the 3# target plate cannot draw a conclusion because of the large difference in areal density. Therefore, supplementary experiments were carried out on target plates with weft densities of 33 picks/cm and 36 picks/cm using the Class I standard in GA141-2010. The ballistic test results are shown in Table 4, and the damage morphology is shown in Figure 17.

It can be seen from Table 4 and Figure 16 that the sag depth of the 4# target plate is smaller than that of the 5# target plate, and the threshold value of the 4# target plate (86%) is higher than that of the 5# target plate (71%). The target plate with a weft density of 33 picks/cm has better trauma resistance than that with a weft density of 36 picks/cm. The reason is that the fabric in the target plate with a weft density of 36 picks/cm is too tight, resulting in a large overall modulus of the target plate and poor deformation ability. When the bullet enters the erosion stage, a large tensile stress is generated in the target plate. The back surface absorbs the remaining energy of the bullet by generating a large deformation. Excessive modulus hinders the ability of the material to deform and absorb energy, which leads to the extension of the bullet penetration distance and the increase in fiber breakage.

In conclusion, the target plate with a weft density of 33 picks/cm can have both lower surface density and better bulletproof performance.

#### 3.2.2. Influence of Layup Methods on Ballistic Resistance

Designing unidirectional layering and orthogonal layering to prepare 3D armor materials. Ballistic test results are depicted in Table 5.

It can be seen from Table 5 that the material back convexity of the material with unilateral layup is smaller while BFS of the material with orthogonal layup is smaller. Unidirectional layup has good resilience, and orthogonal layup is relatively stiff. This is because the modulus of each layer of the unidirectionally layered target plate is similar in all directions, and the deformation of the panel where each layer of fabric is located is similar in each direction when subjected to impact load; the modulus of each layer of the cross-layered target plate in each direction is similar. The moduli in each direction of the orthogonally layered target plate were transposed. It causes each layer of fabric to be constrained by the next layer of fabric during impact deformation, so the material overall exhibits higher stiffness.

The damage morphologies of the 1# and 4# target plates are exhibited in Figure 18. It is clearly seen that bulges and oval white areas on the back surface of the 1# target plate, and the damage areas in the warp and latitude are different indicated by the white arrows in Figure 18a. These phenomena are attributed to the greater degree of stretch in the warp direction than in the weft direction. The warp narrow strips appeared at the 0° shooting position of the back surface of the 6# target plate, which was attributed to the rupture of the weft yarn with poor deformation ability of the back surface after the shock wave.

In conclusion, the unidirectional layup of 3DPPAI armor material has better ballistic performance.

## 4. Conclusions

In this paper, the effects of weft density and layering method on the bending performance and ballistic performance of three-dimensional ply-to-ply angle interlock Kevlar/EP armor materials were studied. Kevlar fabric with three kinds of weft density was prepared. Kevlar/EP armor material was prepared by changing the number of fabric layers and layering method. The bending and bulletproof properties of the armor materials were evaluated. As a result of the experimental investigation presented, the most important conclusions are:1.The bending performance of 3DPPAI Kevlar/EP armor material in the weft direction is obviously better than that in the warp direction. With the increase in the weft density, the warp bending strength and modulus decreased, and the weft bending strength first increased and then decreased;2.The 3DPPAI Kevlar/EP armor material prepared by orthogonal layup has more advantages in bending performance, while the unidirectional layup has better anti-ballistic performance;3.Within the scope of this study, the target plate with a weft density of 33 picks/cm has both excellent bending performance and ballistic resistance.

## Figures and Tables

**Figure 1 materials-15-06994-f001:**
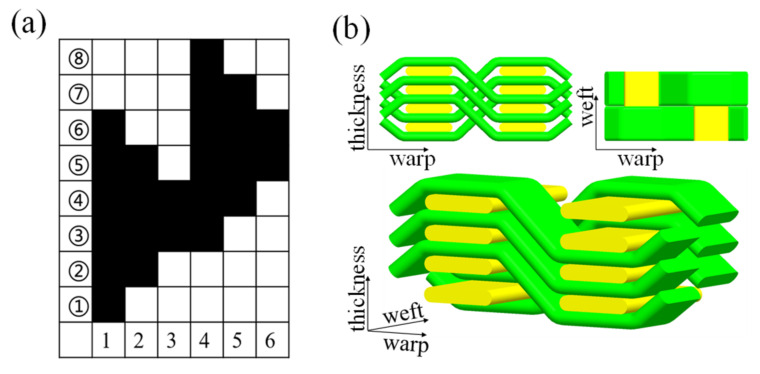
Weave structure design process for 3DPPAI fabric design: (**a**) weave design; (**b**) 3D graphical representation of the woven structure.

**Figure 2 materials-15-06994-f002:**
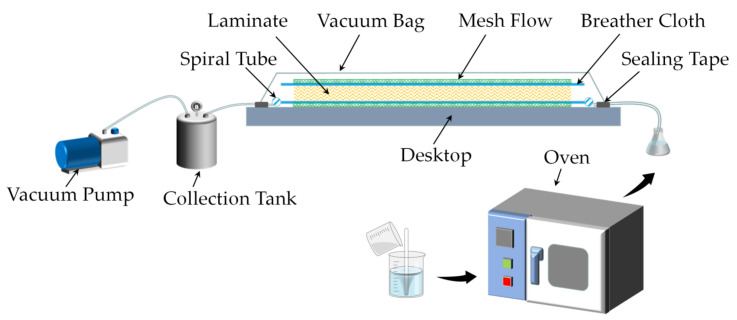
Flow chart of the vacuum-assisted forming process.

**Figure 3 materials-15-06994-f003:**
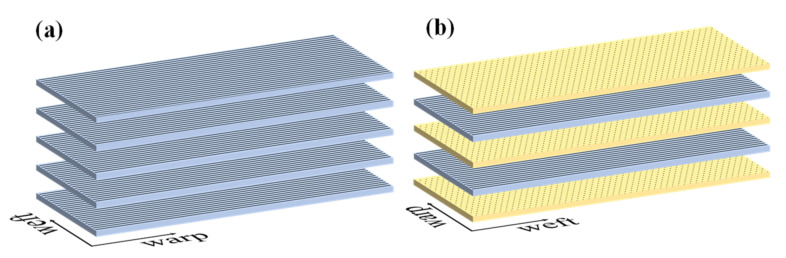
Fabric laying modes: (**a**) Unidirectional ply [0]_n_; (**b**) Orthogonal layup [0/90] _n_.

**Figure 4 materials-15-06994-f004:**
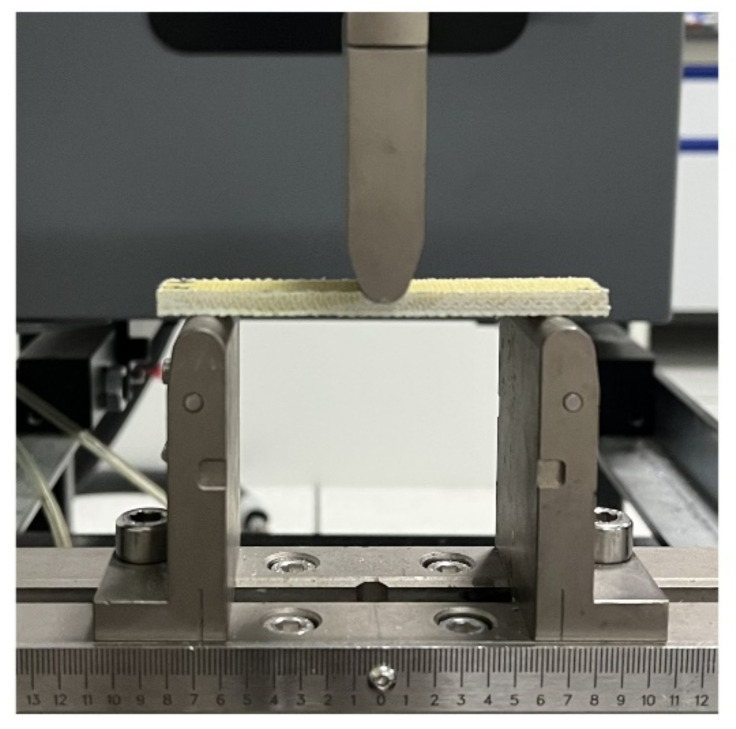
Schematic diagram of three point bending tests.

**Figure 5 materials-15-06994-f005:**
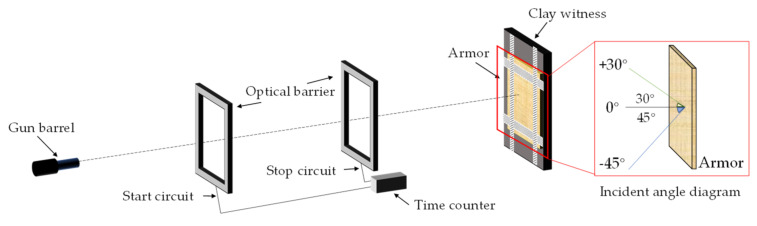
Ballistic experiment setup.

**Figure 6 materials-15-06994-f006:**
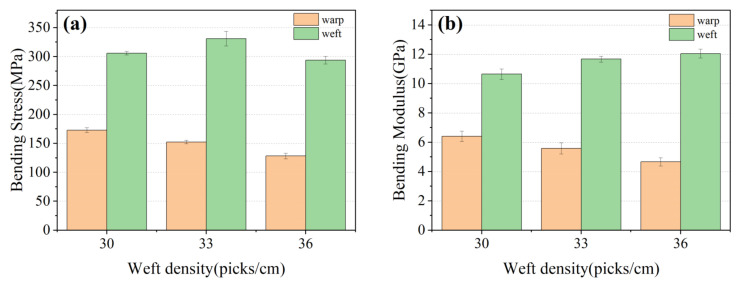
Bending property of 3D armor materials with different weft densities: (**a**) bending strength of the material; (**b**) material flexural modulus.

**Figure 7 materials-15-06994-f007:**
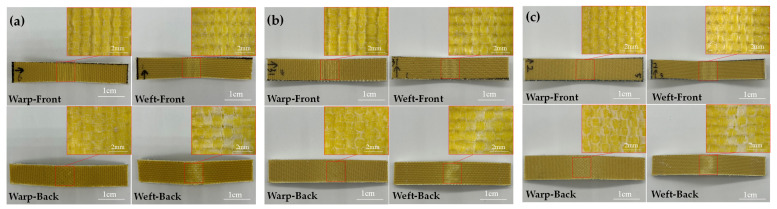
Destruction morphologies of 3D armor materials under different weft densities: (**a**) 30 picks/cm, (**b**) 33 picks/cm, and (**c**) 36 picks/cm.

**Figure 8 materials-15-06994-f008:**
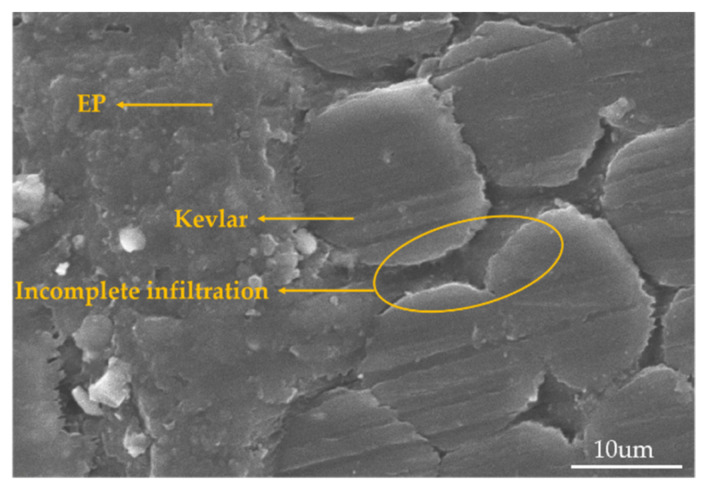
SEM morphology of the cross-section of 3D armor material.

**Figure 9 materials-15-06994-f009:**
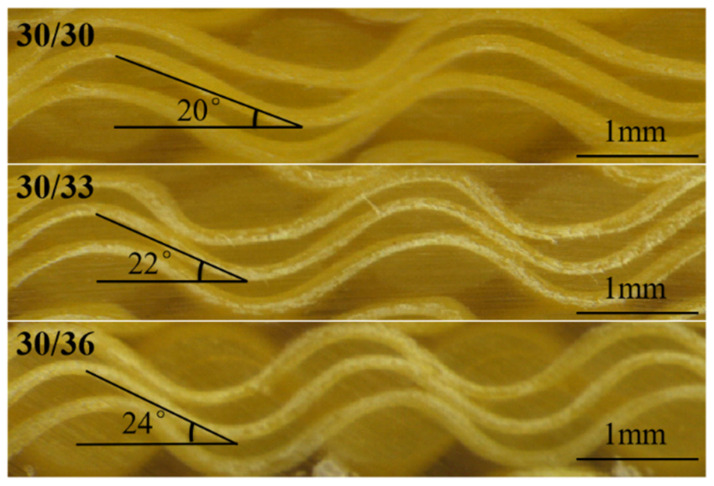
The shape of warp in different weft densities.

**Figure 10 materials-15-06994-f010:**
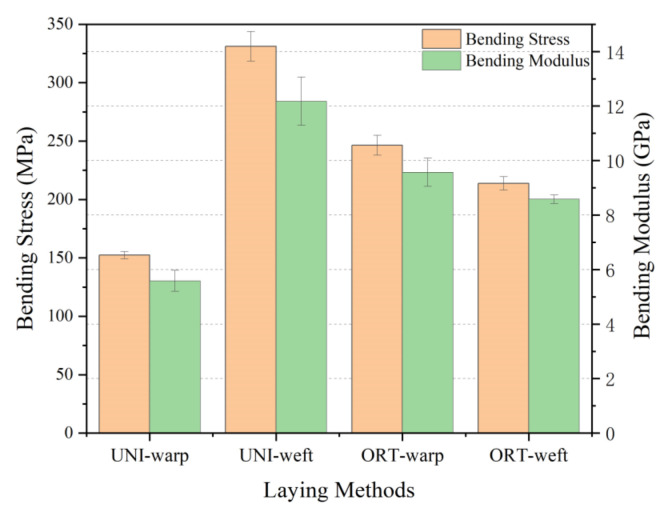
Bending properties of 3D armor materials under different layup methods.

**Figure 11 materials-15-06994-f011:**
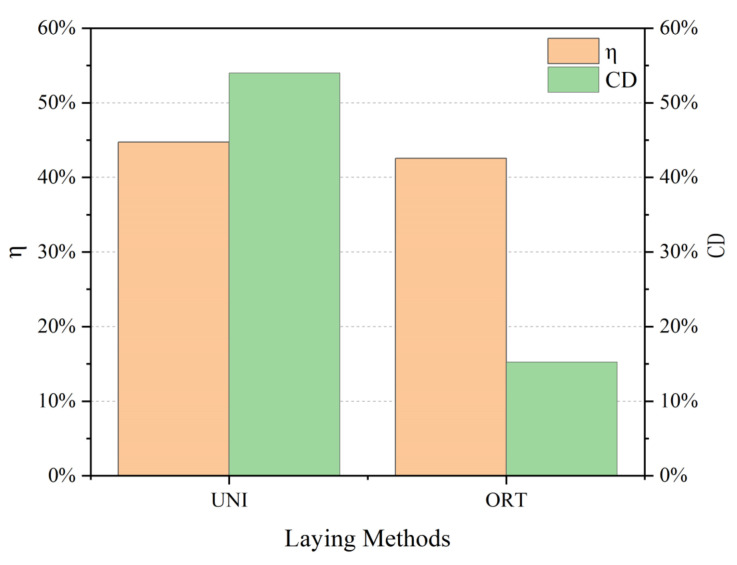
Bending coefficients of 3D armor material under different laying modes.

**Figure 12 materials-15-06994-f012:**
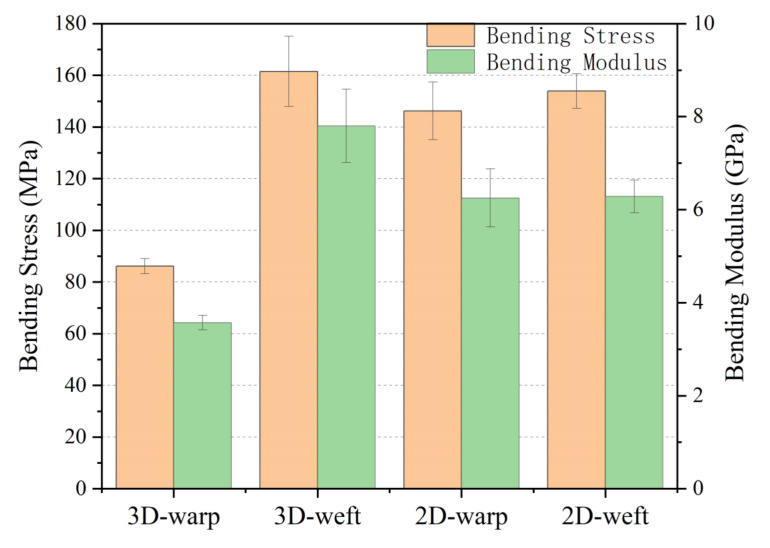
Comparison of bending properties of 3DPPAI and 2D plain weave structures.

**Figure 13 materials-15-06994-f013:**
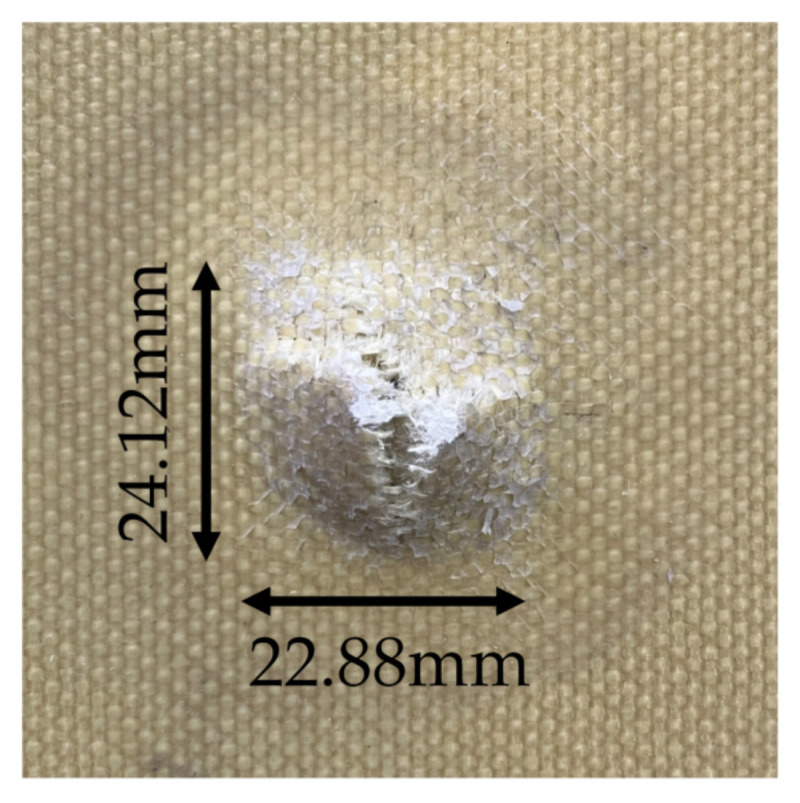
The details of the perforation.

**Figure 14 materials-15-06994-f014:**
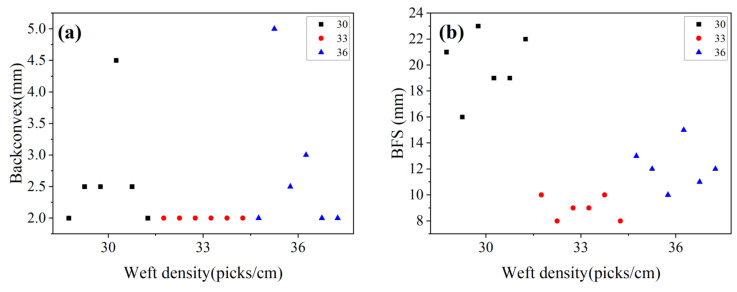
The back convexity and BFS of three kinds of weft densities: (**a**) Back convexity; (**b**) BFS.

**Figure 15 materials-15-06994-f015:**
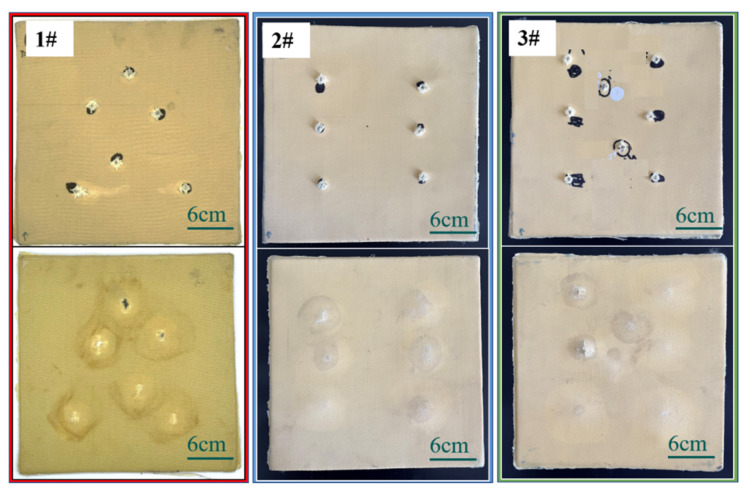
Destruction morphologies of the front and back surfaces of the target plate prepared by fabrics with different weft densities.

**Figure 16 materials-15-06994-f016:**
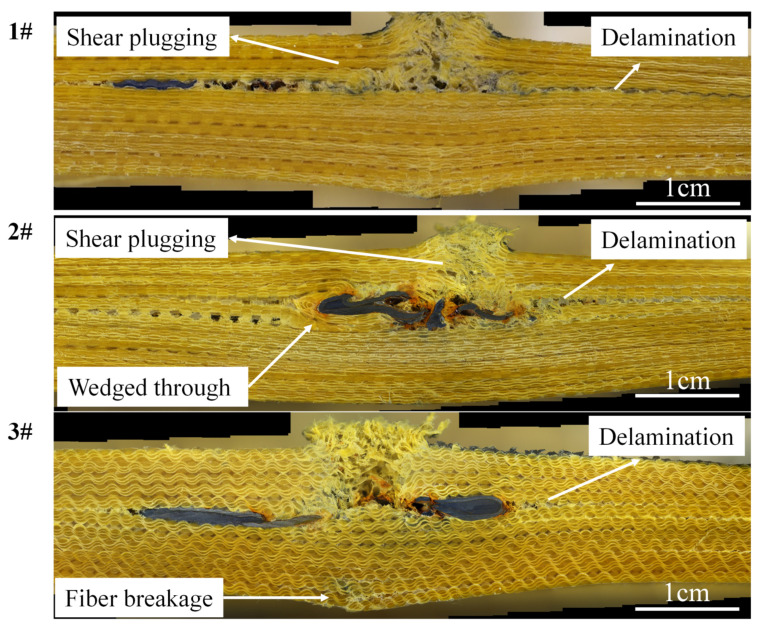
Cross-sectional views of three kinds of armor materials with weft density specifications at 0° firing direction.

**Figure 17 materials-15-06994-f017:**
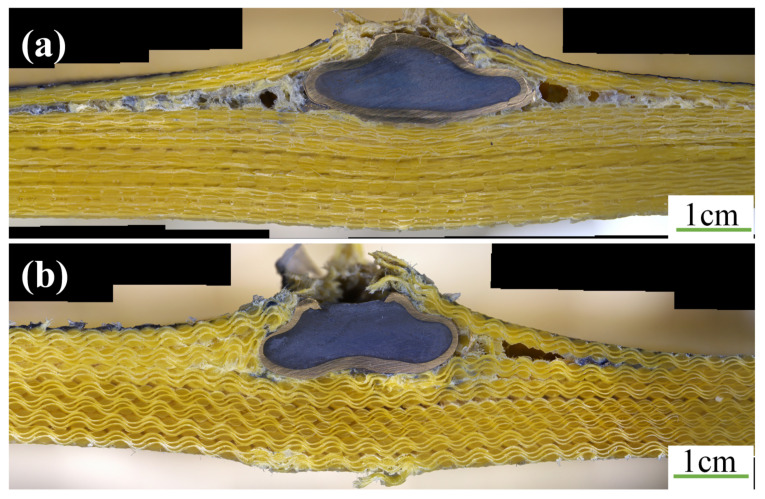
Cross-sectional views of the target plate failure morphology under Class I standard: (**a**) 33 picks/cm; (**b**) 36 picks/cm.

**Figure 18 materials-15-06994-f018:**
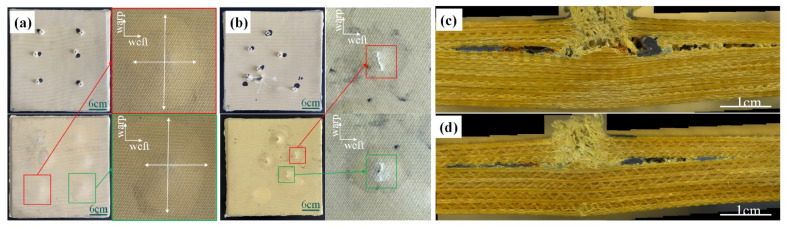
Destruction morphologies of the target plate under different layup methods: (**a**) unilateral ply, (**b**) orthogonal layup; (**c**) cross-sectional view of unilateral ply; and (**d**) cross-sectional view of orthogonal ply.

**Table 1 materials-15-06994-t001:** Specific parameters of 3DPPAI Kevlar fabric.

	Sample Preforms		
Parameters	3DS_1_	3DS_2_	3DS_3_
The total density of warp yarns (ends/cm)	30	30	30
The total density of weft yarns (picks/cm)	30	33	36
No. of warp layers	5	5	5
No. of weft layers	4	4	4
Fabric areal density(g/m^2^)	730	770	810
Fabric thickness (mm)	0.15	0.16	0.18

**Table 2 materials-15-06994-t002:** Specifications of 3DPPAI armor materials and 2D plain Kevlar/EP layered armor materials.

Type	Areal Density/g·m^−2^	Fabric Areal Density/g·m^−2^	Glue Content/%	Thickness/mm
3DPPAI (1 piece)	1.45	810	43.96	1.32
2D plain (4 pieces)	1.41	800	42.89	1.44

**Table 3 materials-15-06994-t003:** Ballistic test results of 3D armor materials with different weft densities.

Target Plate Number	Fabric Specification	Layup Methods	Target Surface Density/kg·m^−2^	Glue Content	Incidence Angle	Velocity/m·s^−1^
1#	30/30	[0]_12_	14.66	41.01%	0°	446
					+30°	445
					−45°	446
					0°	447
					0°	452
					0°	447
2#	30/33	[0]_12_	15.38	40.09%	0°	437
					0°	441
					0°	438
					0°	450
					0°	439
					0°	449
3#	30/36	[0]_11_	14.75	39.22%	0°	451
					+30°	453
					−45°	442
					0°	447
					0°	439
					0°	442

**Table 4 materials-15-06994-t004:** Ballistic test results of 3DPPAI Kevlar/EP armor materials of class I.

Target Plate Number	Fabric Specification	Target Surface Density/kg·m^−2^	Number of Pieces	Glue Content	Incidence Angle	Velocity/m·s^−1^	BFS/mm
4#	30/33	9.11	7	40.73	0°	328	9
					+30°	330	6
					−45°	342	10
					0°	322	9
					0°	327	8
					0°	328	8
5#	30/36	9.25	7	38.90	0°	324	12
					+30°	329	9
					−45°	326	6
					0°	321	13
					0°	327	10
					0°	322	12

**Table 5 materials-15-06994-t005:** Ballistic test results of 3D armor materials under different layup methods.

Target Plate Number	Fabric Specification	Layup Methods	Target Surface Density/kg·m^−2^	Glue Content	Incidence Angle	Velocity/m·s^−1^	BFS/mm	Back Convexity/mm
1#	30/30	[0]_13_	16.24	41.46%	0°	452	10	2
					0°	447	12	2
					0°	448	11	3
					0°	438	12	2
					0°	438	13	2
					0°	444	14	2
6#	30/30	[0/90]_13_	16.24	41.59%	0°	451	12	4
					+30°	439	8	1
					−45°	445	8	1
					0°	446	12	3
					0°	436	12	2
					0°	443	10	2

## Data Availability

Not applicable.

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
