# Peer review of "Research on Bending and Ballistic Performance of Three-Dimensional Ply-to-Ply Angle Interlock Kevlar/EP Armor Material"

_materials, 2022, doi:10.3390/ma15196994_

Round 1

Reviewer 1 Report

Information for Authors

The article submitted for review entitled "Research on Bending and Ballistic Performance of  Three-Dimensional ply-to-ply angle interlock Kevlar / EP Armor Material "raises the problem of the strength of materials used in bulletproof vests. It is an interesting and comprehensive publication, in which the authors presented in a synthetic manner the results of their own research on Kevlar / EP armor materials.  

 The findings obtained in the reviewed article on the basis of the experiment are a novelty.

 The results are presented clearly and the conclusions are shortly formulated, highlighting the most important insights from the research.

 I would like to make one remark regarding the content of the article. After analyzing the world's literature on this subject I noticed that the references such 1-3, 5-10, 12,13, 16-18 are not the latest and are over 10 years old. In connection with the above, I propose to extend the bibliography with a few new similar articles related to the materials used in bulletproof vests, which could be used in the introduction chapter:

·       Burian W., et all., Finite element modeling of ballistic inserts containing aramid fabrics under projectile impact conditions – comparison of methods, Composite Structures, 2022, 115752, ISSN 0263-8223, https://doi.org/10.1016/j.compstruct.2022.115752.

·       Zochowski P., Bajkowski M., Grygoruk R., et all., Ballistic Impact Resistance of Bulletproof Vest Inserts Containing Printed Titanium Structures, Metals, 2021, 11(2), pp. 1–23, 225, https://doi.org/10.3390/met11020225.

·       Wisniewski A. , et all., Anti-trauma pads based on non-Newtonian materials for flexible bulletproof inserts, Proceedings - 29th International Symposium on Ballistics, BALLISTICS 2016, Volume 2, Pages 2034 – 2045, ISBN 978-160595317-5.

·       Leonowicz M., et all., Optimization of the material systems with magnetorheological fluids, Proceedings - 28th International Symposium on Ballistics, BALLISTICS 2014, Volume 2, Pages 1602 – 1612, ISBN 978-160595149-2.

Author Response

Thank you for your comments. I've updated the references and extended the bibliography with a few new similar articles related to the materials used in bulletproof vestsextend the bibliography with a few new similar articles related to the materials used in bulletproof vests. For details, see attached. Looking forward to your reply.

Reviewer 2 Report

Dear Authors, it was pleasure to review paper entitled "Research on Bending and Ballistic Performance of Three-Dimensional ply-to-ply angle interlock Kevlar/EP Armor Material". In general, peper presents good level, but I have some remarks, which can improve the text:

1) I strongly recommend to send this paper to good proofreading service - they will make the text more fluent... Moreover some words are unacceptable - e.g. warhead in case of small-arms ammunition (you should use "bullet"/"projectile");

2) Introduction should be extended. Kevlar-based structures are widely used, investigated and described in literature. You should compare your solution with others. Moreover it would be comfortable in the area of further results analysis. You can compare results of bending test and ballistic tests with other Kevlar-based materials (compare the thickness, aerial mass density etc. needed to stop the projectile);

3) You can include the strain-stress plots for investigated materials and compare it with other Kevlar-based materials (e.g. to show its supremacy). You can also use here results of your previous work in this area (e.g. "Research on Bending Performance of Three-Dimensional Deep Angle Interlock Kevlar/EP Armor Material");

4) Figure presenting the structure of target should be inserted. There should be dimensions and configuration of investigated materials, mug etc.; 

5) Were there any full perforations of small fragments in targets? E.g. in target 1 (Fig. 14) some perforations on back side can be seen. 

6) Only one shot at constant value of incidence angle is not sufficient in ballistic protection analysis. There were applied +30 and -45 deg. impact. Is the sign (negative or positive) important? If you changed incident angle in different reference planes - it should be defined in different way. Please, insert the incidence angle definition in target description figure;

7) You could insert - for comparison - one result of investigation with full perforation (for critical number of pieces). The number of pieces should be included in tables describing targets. 

Thank you

Kind regards

Reviewer 

Author Response

Thank you for your comments.

  1. Regarding question 1, I have already edited the article.
  2. Regarding question 2, I have compared my research protocol with other people's research protocols, showing the strengths of this study (see introduction for details).
  3. Regarding question 3, since the surface density of the material has a great influence on the bending performance and anti-ballistic performance, all the comparisons in this paper are carried out on the premise that the surface density is similar. It cannot be compared with the data in "Research on Bending Performance of Three-Dimensional Deep Angle Interlock Kevlar/EP Armor Material", because the material in this experiment has high bending strength and high areal density, which is not comparable.
  4. Regarding question 4, I have added information such as target board size and shooting angle in 2.5 Ballistic performance test.
  5. Regarding question 5, the GA141-2010 ballistic test standard requires that the bulletproof material is effectively hit with no bullet holes or no bullets on the back bullet surface and the back convexity is less than 25mm. In this paper, the comparison of the pros and cons of bulletproof performance is discussed under the premise of preventing bullets. Therefore, there is no complete small fragment perforation in the target. There were not any full perforations of small fragments in targets. Instead of perforations in Figure 14, there were traces of resin fragmentation and fiber breakage.
  6. Regarding question 6, according to the GA141-2010 ballistic test standard, it is necessary to continuously shoot six shots on the target board. The incident angles of these six guns are clearly required, which are 0°, +30°, -45°, 0°, 0°, 0°, 0°. There are no specific requirements for the plus and minus signs, but it is emphasized that the 30° shooting and the 45° shooting are fired along different sides of the target board. I have modified the ballistic penetration schematic to add an explanation of the angles, see Figure 5 for details.
  7. Regarding question seven, I have added the number of layers of fabric in Chart 4.3

Looking forward to your reply.

Reviewer 3 Report

In the present work, a new Kevlar/EP armor material with a 3D structure was studied for application of high-performance fiber in bulletproof composite materials. I recommend a minor revision before publication. In the following, you can find my comments:

·         Your abstract should clearly state the essence of the problem you are addressing, what you did and what you found and recommend. That will help a prospective reader of the abstract to decide if they wish to read the entire article.

·         Section 2 preparation should be changed to the experimental section.

·         Section 2 should include the materials, preparation and characterizations.

·         The display resolution of all figures should be significantly improved.

Author Response

Thank you for your comments. Regarding question 1, I have revised the abstract part of the article. Regarding question 2, I have adjusted the structure of the article. Regarding question 3, I have uploaded a higher resolution picture, see the attachment for details. Looking forward to your reply.

Round 2

Reviewer 2 Report

Dear Editors, Dear Authors,

Thank you very much for including my remarks.

Kind regards and good luck in further work

Reviewer